# Whole-Genome Identification and Analysis of Multiple Gene Families Reveal Candidate Genes for Theasaponin Biosynthesis in *Camellia oleifera*

**DOI:** 10.3390/ijms23126393

**Published:** 2022-06-07

**Authors:** Liying Yang, Yiyang Gu, Junqin Zhou, Ping Yuan, Nan Jiang, Zelong Wu, Xiaofeng Tan

**Affiliations:** 1Key Laboratory of Cultivation and Protection for Non-Wood Forest Trees, Ministry of Education, Central South University of Forestry and Technology, Changsha 410004, China; ypsnyz05@163.com (L.Y.); gyy11041996@163.com (Y.G.); wuzelong2020@163.com (Z.W.); 2Hunan Horticultural Research Institute, Hunan Academy of Agricultural Sciences, Changsha 410125, China; yuanping@hunaas.cn; 3School of Packing and Material Engineering, Hunan University of Technology, Zhuzhou 412000, China; namijiangnan@126.com

**Keywords:** *Camellia oleifera*, theasaponin, triterpenoid saponin, biosynthesis, regulation

## Abstract

*Camellia oleifera* is an economically important oilseed tree. Seed meals of *C. oleifera* have a long history of use as biocontrol agents in shrimp farming and as cleaning agents in peoples’ daily lives due to the presence of theasaponins, the triterpene saponins from the genus *Camellia*. To characterize the biosynthetic pathway of theasaponins in *C. oleifera*, members of gene families involved in triterpenoid biosynthetic pathways were identified and subjected to phylogenetic analysis with corresponding members in *Arabidopsis thaliana*, *Camellia sinensis*, *Actinidia chinensis*, *Panax ginseng*, and *Medicago truncatula.* In total, 143 triterpenoid backbone biosynthetic genes, 1169 CYP450s, and 1019 UGTs were identified in *C. oleifera*. The expression profiles of triterpenoid backbone biosynthetic genes were analyzed in different tissue and seed developmental stages of *C. oleifera*. The results suggested that MVA is the main pathway for triterpenoid backbone biosynthesis. Moreover, the candidate genes for theasaponin biosynthesis were identified by WGCNA and qRT-PCR analysis; these included 11 *CYP450s*, 14 *UGTs*, and eight transcription factors. Our results provide valuable information for further research investigating the biosynthetic and regulatory network of theasaponins.

## 1. Introduction

*Camellia oleifera*, also called the Camellia oil tree, is a traditionally cultivated woody species in China that is used as a source of high-quality seed oil. Seed meals of *C.**oleifera* have a long history of application along with camellia oil. They have been used as a biocontrol agent in shrimp farming and agricultural production, as a cleaning agent in peoples’ daily lives, and as a protectant to maintain the health of hair. The seed meals of *C.*
*oleifera* contain significant quantities of triterpene saponins, also known as theasaponins. Many studies have suggested that theasaponins have a variety of biological and pharmacological activities, including inhibiting the growth of human carcinoma cells [1,2,3,4], anti-microbial effects [5], anti-inflammatory activity [6], neuroprotection [7], and foaming and detergent properties [8].

Secondary metabolites of natural plants such as triterpene saponins, phytosterols, and flavonoids are vital resources for humans. Identification of the relevant biosynthetic pathways would thus be helpful for our understanding and utilization of theses natural products. To date, more than 70 different theasaponins have been isolated from *Camellia* seeds, all of which are oleanane-type triterpene saponins [9]. However, the biosynthetic pathway of theasaponins is still unclear. In plants, triterpenoid saponins are synthesized from isopentenyl pyrophosphate (IPP) and dimethylallyl diphosphate (DMAPP) derived from the mevalonate (MVA) pathway and methylerythritol phosphate (MEP) pathway. IPP and DMAPP are then transformed into 2,3-oxidosqualene, the common precursor of triterpenoid saponins and sterols in eukaryotes, by the action of farnesyl diphosphate synthase (FPS), squalene synthase (SS), and squalene epoxidase (SE). The above steps are also known as the conserved biosynthesis pathway of the terpenoid backbone. Next, the structural diversity of saponins is formed by three main steps: (1) oxidosqualene cyclases (OSCs) catalyze cyclization of the precursor to different triterpenoid backbones; (2) triterpenoid backbones are modified by cytochrome P450 monooxygenase enzymes (CYP450s); (3) addition of sugar (chains) is catalyzed by UDP-glycosyltransferases (UGTs) [10,11,12].

OSCs catalyze the first committed step in the triterpenoid biosynthesis pathway to generate diverse triterpenoid backbones such as β-amyrin, dammarane, lupine, friedelin, and cycloartenol [13,14,15]. In general, higher plants have several OSCs, including sterol biosynthesis related OSCs such as cycloartenol synthase (CAS) and lanosterol synthase (LAS), and triterpenoid biosynthesis related OSCs such as β-amyrin synthase (bAS) and dammarendiol synthases (DDS) [16,17]. Theasaponins from the genus *Camellia* are oleanane-type triterpenoids [9]. Oleanane-type triterpenoids are widely distributed pentacyclic triterpenoids in the plant kingdom derived from β-amyrin that is generated by bAS [10,18]. Hence, bAS is the key enzyme in the metabolic pathway of theasaponin biosynthesis in *C. oleifera*.

Genes functioning in common processes always have similar expression patterns. Weighted gene co-expression network analysis (WGCNA) is a systems biology approach for describing the co-expression networks between genes across large-scale gene expression profiling data [19,20]. This is a powerful tool for screening potential genes related to biosynthesis and regulation of plant secondary metabolism. *CaMYB48* as a regulator of capsaicinoid biosynthesis was screened by WGCNA [21]. *TSAR1*, *TSAR2*, multiple CYP450s, and UGTs were predicted to be involved in triterpene saponin biosynthesis in *Medicago truncatula* through co-expression analysis [22,23]. These examples show that candidate genes for theasaponin biosynthesis in *C. oleifera* could be revealed by WGCNA.

In this study, we selected genes involved in triterpene saponin biosynthetic pathways in *C. oleifera*, including MVA and MEP pathways and the IDI, FPS, SS, SE, OSC, CYP450, and UGT families, and constructed a gene co-expression network by WGCNA based on the expression pattern of *bAS*. The resulting network was combined with correlation results obtained through quantitative real-time PCR (qRT-PCR) to screen for candidate genes involved in theasaponin biosynthesis. This study establishes a foundation for further research investigating the biosynthetic and regulatory networks of theasaponins.

## 2. Results

### 2.1. Identification of Triterpenoid Saponin Biosynthesis Related Genes

To identify genes involved in theasaponin biosynthesis in *Camellia oleifera*, we explored the predicted protein database of *C. oleifera* ‘Huashuo’ and five other species (*A. thaliana*, *C. sinensis*, *A. chinensis*, *P. ginseng*, and *M. truncatula*) by an HMM (hidden Markov model) search. Based on conserved domain (Appendix A) and phylogenetic analyses (Appendix A), 452 putative triterpenoid backbone biosynthetic genes, including MVA and MEP pathways, IDI, FPS, SS, SE, and OSC families, were identified. Additionally, 2909 CYP450s and 1986 UG3Ts were identified in the six species. Among those genes, there were 143 triterpenoid backbone biosynthetic genes, 1169 CYP450s, and 1019 UGTs from *C. oleifera* (Table 1). The numbers of CYP450s and UGTs in *C. oleifera* were much greater than those in the other five species.

CYP450 is one of the largest gene families in plants and is often recruited as a versatile catalyst in the biosynthesis of plant specialized compounds. To perform a detailed classification of CYP450s, a phylogenetic tree was constructed with the 2909 CYP450s identified above (Table 2, Appendix A). As a result, 2897 genes were assigned into 9 CYP450 clans comprising 46 families, and 12 other CYP450s (3 from *C. oleifera*, 1 from *A. thaliana*, 5 from *A. chinensis*, 2 from *P. ginseng*, and 1 from *M. truncatula*) were not classified into any of the above families. The CYP71 clan was the largest with 19 families, representing all A-type CYP450s. Except for *A. chinensis* (33.5%), the members of the CYP71 clan accounted for more than half of all CYP450s. In *C. oleifera*, 41 families contained members, while 5 families (CYP702, CYP705, CYP708, CYP709, and CYP712) contained no members. In addition, the CYP71 family was the largest. CYP716, CYP72, CYP88, and CYP93 families contained 65, 97, 12, and 6 members, respectively, in *C. oleifera*; these families had been characterized as associated with triterpenoid saponin modification in other species. In addition, the members of CYP79 (55 in *C. oleifera*, 10 in *A. thaliana*, 1 in *C. sinensis*, 3 in *A. chinensis*, 6 in *P. ginseng*, 7 in *M. truncatula*), CYP82 (102 in *C. oleifera*, 5 in *A. thaliana*, 28 in *C. sinensis*, 9 in *A. chinensis*, 20 in *P. ginseng*, and 19 in *M. truncatula*), and CYP87 (57 in *C. oleifera*, 1 in *A. thaliana*, 9 in *C. sinensis*, 9 in *A. chinensis*, 8 in *P. ginseng*, and 1 in *M. truncatula*) in *C. oleifera* were far greater than those in the five other species, and the members of CYP83 in *C. oleifera* (21), *P. ginseng* (12), and *M. truncatula* (19) were far greater than those in *A. thaliana* (2), *C. sinensis* (2), and *A. chinensis* (1).

In plants, UGTs are responsible for transferring glycosyl moieties to acceptor molecules, including theasaponins. We identified 1986 UGTs in the six species, with 1019 UGTs from *C. oleifera*, accounting for 51.31% of the total UGTs. The lengths of these putative UGTs of *C. oleifera* were 127–1273 amino acids. Those identified UGTs were aligned with three functionally characterized UGTs from *Zea mays* to construct a phylogenetic tree. As a result, those genes classified into 25 families belonged to 16 groups, including A-N (the conserved groups that were identified in *Arabidopsis*), and O, P groups (two novel groups identified in maize) while no UGTs were phylogenetically separated into the Q group (Table 3, Appendix A) [24,25]. The A group was the most abundant *C. oleifera* UGT group, containing UGT79, UGT80, UGT91, and UGT94 gene families, followed by the E group and the L group. UGT71, UGT73, UGT74, and UGT94 families were characterized as triterpene glucosyltransferases. These families encompassed a large number of members of *C. oleifera* UGT, containing 60, 110, 76, and 99 members, respectively.

### 2.2. Expression of Triterpenoid Backbone Biosynthetic Genes in C. oleifera

In this study, a total of 143 triterpenoid backbone biosynthetic genes were identified in *C. oleifera* (Table 1). Of these, 50 genes encoded six key enzymes involved in the MVA pathway, including acetyl-CoA C-acetyltransferase (AACT, 7 genes), 3-hydroxy-3-methylglutaryl-CoA synthase (HMGS, 6 genes), 3-hydroxy-3-methylglutaryl-CoA reductase (HMGR, 18 genes), phosphomevalonate kinase (MVK, 4 genes), mevalonate kinase (PMK, 6 genes), and mevalonate diphosphate decarboxylase (MVD, 9 genes). A total of 39 genes encoded seven key enzymes of the MEP pathway, including 1-deoxy-D-xylulose-5-phosphate synthase (DXS, 13 genes), 1-deoxy-D-xylulose-5-phosphate reductase (DXR, 6 genes), 2-C-methyl-D-erythritol 4-phosphate cytidyltransferase (MCT, 3 genes), 4-(cytidine 5′-diphospho)-2-C-methyl-D-erythritol kinase (CMK, 8 genes), 2-C-methyl-D-erythritol 2,4-cyclodiphosphate synthase (MDS, 5 genes), 1-hydroxy-2-methyl-2-butenyl 4-diphosphate synthase (HDS, 3 genes), and 1-hydroxy-2-methyl-2-butenyl 4-diphosphate reductase (HDR, 1 gene). A total of 23 genes encoded four putative enzymes (IDI, FPS, SS and SE) that were found to be associated with conversion of IPP to 2,3-oxidosqualene. Additionally, 31 genes encoded the oxidosqualene cyclase (OSC) that cyclizes 2,3-oxidosqualene to generate sterol or triterpenoid.

OSC catalyzes the first committed step in the triterpenoid biosynthesis pathway, and it plays an important role in the formation of diverse triterpenoid backbones. It has been found that OSCs such as CAS and LAS catalyze the generation of sterols, while bAS and DDS catalyze the generation of triterpenoids. In this work, a total of 83 OSCs were identified, 31 from *C. oleifera*. All of the OSCs contained two conserved domains, SQHop_cyclase_C and SQHop_cyclase_N. Those OSCs and some functionally characterized OSCs were aligned and used to construct a phylogenetic tree (Figure 1A). The results showed that *C. oleifera* OSCs consisted of two sterol-related OSCs (1 CAS and 1 LAS) and 29 triterpenoid-related OSCs (16 bASs and 13 DDSs).

The transcriptomes of different tissues (root, stem, leaf, petal, filament, anther, style, and ovary) and different seed developmental stages (seeds in July, August, September, and October) of *C. oleifera* were used for the expression analysis of triterpenoid backbone biosynthetic genes. The *OSC* genes displayed divergent expression patterns (Figure 1B). In sterol-related *OSCs*, *CAS* (*Co10022113*) was expressed equally in all samples; *LAS* (*Co10411291*) showed a low level of expression in roots. For triterpenoid-related *OSCs*, six *bAS* genes (*Co10176456*, *Co10051732*, *Co10352372*, *Co10286845*, *Co10307396*, and *Co10350615*) and two *DDS* genes (*Co10339102* and *Co10389481*) showed high expression in some samples, while other *OSC* genes were not expressed or were expressed at very low levels. The six *bASs* were strongly expressed in roots, stems, leaves, and especially in the seeds in August, September, and October. The two *DDSs* showed high levels of expression in roots and stems, especially in roots. In addition, no triterpenoid-related *OSCs* were expressed in any flower parts, except for three *bASs* and two *DDSs* that were expressed at low levels in filaments and styles, respectively. Those results indicated that triterpenoids are primarily derived from the β-amyrin scaffold in *C. oleifera* seeds, while in roots, stems, and leaves they are derived from β-amyrin and dammarendiol scaffold.

To further screen for major-effect genes and pathways involved in triterpenoid backbone biosynthesis in *C. oleifera*, the gene expression data of identified genes in MVA, MEP, and IPP-related downstream pathways were used for a clustering analysis (Figure 1C, Appendix A). A total of 27 genes were grouped together with the total FPKM of 29 triterpenoid-related *OSC* genes. In other words, the 27 genes exhibited a similar expression pattern as the triterpenoid-related *OSCs*, and thus may be responsible for triterpenoid backbone biosynthesis. Most of these genes (14 genes) were involved in the MVA pathway, and 8 genes were IPP-related downstream genes. Except for MVD, all of the enzymes of MVA and IPP-related downstream pathways were encoded by one or multiple genes in this cluster that contained four AACT genes, four HMGS genes, one HMGR gene, two MVK genes, three PMK genes, two IDI genes, three FPS genes, one SS gene, and two SE genes. This result suggested that MVA was the main pathway for triterpenoid backbone biosynthesis in *C. oleifera*. Moreover, an *HMGR* (*Co10265376*) and an *SE* (*Co10322248*) in this cluster exhibited extremely high expression in August and September in *C. oleifera* seeds, with average FPKM values of 1047, 1011, and 923, 990, respectively (Appendix A). This indicated that the *HMGR* and *SE* genes may play important roles in the theasaponin biosynthetic pathway in *C. oleifera* seeds (Figure 2).

### 2.3. Identification of Candidate CYP450s, UGTs, and Transcription Factors Related to Theasaponin Synthesis in C. oleifera Seeds by WGCNA

CYP450 and UGT are large gene superfamilies, each containing more than 1000 members in *C. oleifera*; this makes it difficult to obtain candidate genes involved in theasaponin biosynthesis. As genes belonging to the same pathway had similar expression patterns in different tissues and developmental periods, we generated a co-expression network via WGCNA using the FPKM values of all above-identified genes and predicted transcription factors in *C. oleifera* as source data. As the enzyme bAS catalyzes the first and most critical step of the theasaponin biosynthesis pathway in *C. oleifera* seeds, we considered the FPKM of *bASs* as representative of the level of theasaponin biosynthesis. Finally, among 11 modules, the MEbrown module containing 475 genes had an expression pattern tightly correlated with theasaponin biosynthesis (Figure 3A,B). The heatmap of MEbrown genes (Figure 3C) demonstrated that most of these genes exhibited a seed-specific pattern and had a higher expression level in seeds in August, September, and October. There were 291 genes with gene significance (GS) > 0.7 and intramodular connectivity (kME) values > 0.7 in this module. Consistent with the results of the clustering analysis of triterpenoid backbone biosynthetic genes, there were several genes for encoding enzymes of MVA and IPP-related downstream pathways among these 291 genes. Additionally, 41 *CYP450s* and 40 *UGTs* were present and may be related to theasaponin biosynthesis (Appendix A).

Apart from the above, fifteen transcription factors (TFs) with GS > 0.8 and kME > 0.95 in the MEbrown module, in which three basic helix-loop-helix transcription factors (*bHLH*), four growth-regulating factors (*GRF*), four B3 domain-contain transcription factors (*B3*), and one lateral organ boundary domain gene (*LBD*) were present. In addition, eight *MYB* transcription factors, an important gene family in regulation of the biosynthesis of secondary metabolites, were contained in the MEbrown module with kME > 0.90 (Figure 3D, Appendix A).

### 2.4. Verification of the Results in WGCNA by qRT-PCR

To further screen for candidate CYP450s, UGTs, and TFs involved in theasaponin biosynthesis in *C. oleifera* seeds, we first aligned CDSs of CYP450s, UGTs, and TFs from WGCNA results and *bASs*. Genes with identity ≥ 96% were regarded as alleles. In the results, 17 non-allelic *CYP450s*, 25 non-allelic *UGTs*, three *bHLHs*, three *GRFs*, three *B3s*, one *LBD* gene, seven *MYBs*, and one *bAS* were obtained and named according to the family they belonged to (Appendix A). Relative expression levels of these genes were then quantified by qRT-PCR in six different developmental stages (30 June, 15 July, 30 July, 30 August, 30 September, and 20 October) of *C. oleifera* seeds (Figure 4A). The results were in general agreement with those from the RNA-Seq; most of those genes had relatively high-level expression in August, September, and October, while having relatively low-level expression in June and July (Figure 4B). Finally, the Pearson correlation coefficient (R) was calculated between each selected gene and *bAS* using their relative expression (Appendix A). Genes with R < 0.7 were dropped. Finally, we obtained 11 *CYP450s* (three in *CYP71*, two in *CYP716*, and one each in *CYP72*, *CYP73*, *CYP79*, *CYP81*, *CYP83*, and *CYP87*) (Figure 4C), 14 *UGTs* (two each in *UGT73*, *UGT91*, and *UGT93* and one each in *UGT72*, *UGT75*, *UGT78*, *UGT79*, *UGT80*, *UGT90*, *UGT94*, and *UGT708*) (Figure 5) and eight TFs (two in *bHLH*, one in *B3*, one in *GRF*, and four in *MYB*) (Figure 6) as the candidate genes indicating a variety of structures and complex mechanisms involved in theasaponin biosynthesis and regulation in *C. oleifera* seeds. Among the 11 *CYP450s* and 14 *UGTs*, *CoCYP716-1*, *CoCYP716-2*, *CoCYP87-1*, and *CoUGT73-1* exhibited extremely high FPKM values (Appendix A) and relative expression (Figure 4 and Figure 5) in *C. oleifera* seeds during August, September, and October.

## 3. Discussion

### 3.1. Biosynthesis Pathway of the Triterpenoid Backbone in C. oleifera

In addition to high-quality edible oils, the *C. oleifera* seeds contain abundant secondary metabolites such as flavonoids, saponins, phytosterols, and squalenes. Saponin is one of the main active ingredients extracted from *Camellia oleifera* seeds. In the past several decades, theasaponins, a group of triterpenoid saponins from the genus *Camellia*, have received increasing attention due to their bioactivities. Many individual theasaponins have been isolated, structurally characterized, and functionally identified [3,9,26]. However, the biosynthetic pathway of theasaponin in *C. oleifera* has not yet been completely resolved. As a widespread bioactive compound in plants, triterpenoid saponin has been studied extensively not only in regard to structure and function but also concerning the biosynthetic pathway in many species, especially in medicinal plants such as *Panax ginseng* [27,28], *Bupleurum falcatum* [29,30], *Platycodon grandiflorus* [12], and model plants such as *Medicago truncatula* [23,31]. Generally, the triterpenoid saponins synthesis pathway can be divided into two stages, the triterpenoid skeleton synthesis stage and the triterpenoid skeleton modification stage. Triterpenoid skeleton modification includes oxidation modification catalyzed by CYP450, glycosylation modification catalyzed by UGT, and other modifications.

In this study, we mined members of the gene families related to triterpenoid saponin synthesis, and most of those families had more members in *C. oleifera* than in the other five species, especially the *CYP450* and *UGT* gene families. This may be because there are more alleles in *C.*
*oleifera* because it is a hexaploid and is highly polymorphic. Further analysis of the expression of triterpenoid skeleton biosynthetic genes revealed that genes presenting similar expression patterns as triterpenoid-related *OSCs* were concentrated in the MVA pathway, while the MEP pathway was less prominent. We speculated that the MVA pathway was the main pathway for triterpenoid backbone biosynthesis in *C. oleifera*. This result was consistent with previous studies [16]. HMGR is a rate limiting enzyme in the MVA pathway, and 18 *HMGRs* were identified in *C. oleifera* and exhibited diverse expression patterns. One of the eighteen *HMGRs* had a very similar expression pattern to *bAS* in different seed developmental stages of *C. oleifera* and in addition had a very high expression level. As in HMGR, one of the eleven SE genes also had a similar expression pattern and high expression level in *C. oleifera* seeds. We speculated that these two genes play important roles in theasaponin biosynthesis, and different genes from the same family may be responsible for different end-products of biosynthesis. OSCs catalyze the first step in the specific biosynthesis of triterpenoid saponins and determine the structure of diverse triterpene skeletons. *OSCs* expression analysis showed that only *bAS* exhibited high-level expression in *C. oleifera* seeds. This finding indicates that the triterpenoids are primarily derived from the β-amyrin scaffold in *C. oleifera* seeds, explaining why all the theasaponins detected in *C. oleifera* seeds to date are oleanane-type triterpene saponins [9]. The expression trends of *bAS* indicated that rapid saponins synthesis in *C. oleifera* seeds occurred in August, September, and October. In addition, as in other species, saponins have different accumulation levels and types in different tissue parts and developmental stages of *C. oleifera*.

### 3.2. Candidate CYP450s Involved in Theasaponin Biosynthesis in C. oleifera Seeds

Extensive experimentation has shown that *C. oleifera* seeds contain a variety of oleanane-type triterpenoids such as oleiferasaponin [32,33,34] and camelliasaponin [33]. A basic structure of theasaponin is shown in Figure 2B,C, illustrating that oxidative modifications exist at C-16, C-21, C-22, C-23, and C-28 of triterpenoid backbones. By far the most common modifications found in triterpene saponins are catalyzed by CYP450s. CYP716 is an ancient CYP450 gene family in the CYP85 clan, and its proposed origin is in triterpenoid primary metabolism [35]. CYP716 enzymes are to date the only CYP450s known to perform C-28 three-step oxidation of triterpenoids [36]. Of all the candidate *CYP450s* obtained in this study, there were two non-allelic *CYP716s*, *CoCYP716-1* and *CoCYP716-2*. In *C. oleifera* seeds, they had similar expression patterns to *bAS*, and extremely high expression in August and September from transcriptome data (Appendix A) and qRT-PCR (Figure 4C). Moreover, *CoCYP716-1* and *CoCYP716-2* exhibited a high-level sequence identity (more than 50%) with *P. ginseng CYP716A52v2* and *Platycodon grandiflorus CYP716A140v2* that are β-amyrin C-28-oxidase enzymes involved in oleanolic acid production [30,37]. As such, we speculated that the C-28 oxidation of β-amyrin in *C. oleifera* seeds is most likely performed by *CoCYP716-1* and *CoCYP716-2*. Nonetheless, most of the characterized CYP716s catalyze C-28, but not only C-28 oxidation. For example, *CYP716A141* from *Platycodon grandifloras* was characterized as a C-16β hydroxylation enzyme [38]; *CYP716Y1* from *Bupleurum falcatum* catalyzed C-16α hydroxylation of β-amyrin [30]; and *CYP716A2* displayed 22α-hydroxylation activity in *Arabidopsis thaliana* [39], and *CYP716A14v2* oxidized the C-3 hydroxyl group to carbonyl group, a prerequisite for further additions such as glycosylation at this position [40]. These results suggest that *CoCYP716-1* and *CoCYP716-2* may also have catalytic activity on C-16, C-22, and C-3 of theasaponin. CYP87D16 is another CYP450 that has C-16α oxidase activity in *Maesa lanceolate* [41]. This enzyme belongs to the CYP87 family that, similar to the CYP716 family, is a member of the CYP85 clan. We screened a member of the CYP87 family (*CoCYP87-1*, containing three alleles, *Co10319325*, *Co10341811*, and *Co10360453*) as a candidate gene involved in theasaponin biosynthesis in *C. oleifera* seeds. *CoCYP87-1* had an amino acid similarity of 76.9% to *CYP87D16*, and thus we hypothesized that *CoCYP87-1* was the best candidate gene for C-16 oxidation of theasaponin in *C. oleifera* seeds.

C-23 is one of the most common positions for oxidation of the triterpenoid backbone, and the presence of the hydroxyl group at C-23 is crucial for biological activity [38]. Previous studies demonstrated that C-23 oxidation of oleanolic acid is catalyzed by *CYP72A68v2* in *Medicago truncatula* [42], *CYP72A552* in *Barbarea vulgaris* [43], and *CYP71A16* in *Arabidopsis thaliana* [44]. In our analysis, three CYP71 genes and one CYP72 gene were co-expressed with *bAS*, suggesting the possibility that one or more of those genes performed C-23 oxidation in theasaponin biosynthesis. Additionally, some saponins contained additions at C-21 and C-22, but the enzyme is largely unknown. In this study, in addition to the above genes, four *CYP450s* belonging to CYP73, CYP79, CYP81, and CYP83 were also co-expressed with *bAS*. This suggested that those genes may have catalytic activity on β-amyrin or related compounds. Nonetheless, CYP450s are ubiquitous enzymes; some from two different gene families exhibit the same biochemical function [41], while some from the same family perform different biochemical functions [42,45]. Our current findings suggest possible yet unexplored functions of candidate *CYP450s* related to theasaponin biosynthesis in *C. oleifera* seeds.

### 3.3. Candidate UGTs Involved in Theasaponin Biosynthesis in C. oleifera Seeds

Glycosylation, which is usually catalyzed by UGTs, can alter bioactivity and solubility and increase the diversity of saponins [46]. In theasaponin, sugar moieties are attached to C-3 with a glucuronic acid (GlcA) or its methyl ester, and substituted at position 2′ (one sugar unit) and position 3′ (one or two sugar units) by glucose (Glc), galactose (Gal), arabinose (Ara), xylose (Xyl), or rhamnose (Rha) (Figure 3C) [9]. Usually, the diversification of triterpenoids created by UGTs has been thought to be by far the most common [47]. However, a few UGT enzymes have been identified to glycosylate triterpene aglycones. Earlier studies showed that UGTs glycosylated triterpenes are mostly members of groups A, D, and E [47]. Group A contains UGT79, UGT80, UGT91, and UGT94 families; group D contains the UGT73 family, and group E contains UGT71, UGT72, UGT88, and UGT708 families (Table 3). In this study, most of the candidate *UGTs* (nine of fourteen) belonged to groups A, D, or E, implying that the results are reliable. The other five genes comprised two *UGT93s*, one *UGT75*, one *UGT78*, and one *UGT90*. These may be undiscovered genes with glycosylated triterpenoids, or they may catalyze other reactions that happen to coincide with the biosynthesis of saponins in *C. oleifera* seeds. In previous studies, several UGT73s have been shown to have the function of glycosylating oleanane-type triterpene saponins at the C-3 position. UGT73C10, UGT73C11, and four OAGTs (members of the UGT73 family) are responsible for the addition of the first sugars of the C-3 sugar chain [48,49], while UGT73P2 (galactosyltransferase) [50] and UGT73P10 (arabinosyltransferase) [51] catalyze the addition of the second sugars of the C-3 sugar chain. In our current study, there were two *UGT73s* as candidate genes. The two genes, especially *CoUGT73-1*, had high FPKM values (Appendix A) and relative expression levels (Figure 5) in *C. oleifera* seeds in August, September, and October. Moreover, amino acid sequences of the two genes were more similar to UGT73P2 (about 45%) than to UGT73C10 (about 40%), suggesting that the two *UGT73s* are more likely to be responsible for the addition of second sugars than first sugars of the C-3 sugar chain. Cellulose synthase is another gene superfamily identified as the first glucuronosyltransferase at the C-3 position of oleanane-type aglycones [52,53]. However, we did not analyze this superfamily in this study. The functions of UGTs are very complicated, and the correlation between substrate selectivity and the sequence is very low, making it difficult to identify the target UGTs. Hence, the candidate *UGTs* we obtained need further in vitro and in vivo functional analysis.

### 3.4. TFs Involved in Theasaponin Biosynthesis in C. oleifera Seeds

Plants deploy a variety of secondary metabolites as defense mechanisms against various stress situations. Their biosynthesis is tightly regulated, and multiple phytohormones, such as jasmonate (JA) and salicylic acid (SA), are involved in the process. Previous studies and this report imply that the triterpene biosynthetic and regulation networks are extremely complex, and many transcription factors participate in the pathway. *bHLH* is the most-reported transcription factor family involved in the regulation of saponin biosynthesis, as *TSAR1-3*, *MYC2* and *TSARL1-2* [54,55,56]. These usually directly bind to the promoters of triterpene biosynthetic genes and could be induced by JAs. *MYB* is a famous transcription factor family that participates in the regulation of secondary metabolite biosynthesis. There is some evidence that *MYB* can regulate triterpene biosynthesis. For example, *BpMYB21* and *PgMYB2* positively regulate triterpenoid biosynthesis, while the *VvMYB5b* gene decreases β-amyrin in tomato [57,58,59]. *MYB* can directly modulate secondary metabolites or form complexes with *bHLH* to regulate secondary metabolites. In this study, two *bHLHs*, one *GRF*, one *B3*, and four *MYBs* exhibited high co-expression with *bAS* (Figure 6), and they may thus regulate the theasaponin biosynthesis in *C. oleifera* seeds. Whether these *CobHLHs* and *CoMYBs* act directly or form complexes to regulate theasaponin biosynthesis in *C. oleifera* seeds remains to be further investigated. The other transcription factors we screened have not been shown to be directly involved in the regulation of triterpenoid synthesis, but they play crucial roles in many important biological processes including secondary metabolites and stress responses. They can also interact with many transcription factors, such as *MYB* and *bHLH*. Thus, we cannot rule out the possibility that they also participate in theasaponin biosynthesis in *C. oleifera* seeds. In conclusion, as for other biological processes, theasaponins biosynthesis involves a complex network with co-functioning of multiple transcription factors and structural genes. Whether the transcription factors we screened actually regulate theasaponin biosynthesis and how to regulate this process merit further study.

## 4. Materials and Methods

### 4.1. Data Resources Used

The hidden Markov model (HMM) files corresponding to the domains were downloaded from the Pfam protein family database (http://pfam.xfam.org/, accessed on 8 April 2021). The domains each family contained and their Pfam IDs are provided in Appendix A.

The genomic data were obtained from the following websites: *Arabidopsis thaliana*, http://plants.ensembl.org/Arabidopsis_thaliana/Info/Index, accessed on 27 May 2021; *Camellia sinensis* [60,61], http://tpia.teaplant.org/download.html, accessed on 23 April 2021; *Actinidia chinensis* [62], http://kiwifruitgenome.org/, accessed on 6 May 2021; *Panax ginseng* [27], http://gigadb.org/dataset/view/id/100348/File_page/3, accessed on 15 April 2021; *Medicago truncatula* [63], https://ftp.ncbi.nlm.nih.gov/genomes/refseq/plant/Medicago_truncatula/latest_assembly_versions/, accessed on 7 July 2021; *Camellia oleifera* ‘Huashuo’, unpublished data from our research group.

RNA sequencing data were retrieved from previous studies by our research group [64], with accession number: PRJNA 693152.

### 4.2. Identification of Triterpenoid Saponin Biosynthesis-Related Genes

According to Appendix A, hmmsearch v.3.1b1 was used to screen all proteins containing these domains from the predicted protein database of *C. oleifera* ‘Huashuo’, *A. thaliana*, *C. sinensis*, *A. chinensis*, *P. ginseng*, and *M. truncatula*, with 1e-3 as the threshold E-value. Proteins that exceeded this threshold were analyzed for the existing domains using the plug-in “Batch SMART” of TBtools [65] (v1.098696). This plug-in links to the SMART website (http://smart.embl-heidelberg.de/, accessed on 12 July 2021). If domains belonged only to the specified family, the domain-containing proteins were identified as members of that family. If not, a phylogenetic tree was constructed, and identified family members were based on known proteins from *A. thaliana* and other species.

### 4.3. Sequence Alignment and Phylogenetic Tree Construction

Amino acid sequences were aligned using MAFFT v7.215. The approximate maximum likelihood tree was constructed by FastTree (v2.1.7 SSE3) using the JTT + CAT model and SH-like test with 1000 resamples [66,67]. The phylogenetic trees were visualized and drawn using MEGA7 and Adobe Illustrator 2020 software (*Adobe* Inc., San Jose, CA, USA).

### 4.4. Visualization of Gene Expression and Identification of Co-Expression Modules

The RNA sequencing data were re-analyzed with the *C. oleifera* ‘Huashuo’ genome as a reference, and the values of FPKM (the fragments per kilobase of exon per million mapped reads) were calculated for each transcript. Gene expression visualization was realized by the plug-in “HeatMap” for TBtools. A gene co-expression network was built using the plug-in “WGCNA shiny” for TBtools. The networks were visualized using Cytoscape v.3.8.2 (Cytoscape Consortium, USA).

### 4.5. Plant Materials

The different developmental stages of seeds of *C. oleifera* ‘Huashuo’ used for RT-qPCR were collected at the HuJu forest farm in Chaling county, Zhuzhou, Hunan Province, China (113° 25′ E, 26° 55′ N). Nine healthy trees in adjacent geographical locations with the same age and the same growth potential were randomly selected, and we marked the flowers that bloomed on the same day in the full-bloom stage. Three trees with the most marked fruits were then selected, and each tree was considered as a biological replicate. Finally, the samples were randomly collected from marked fruits of the three trees on 30 June, 15 July, 30 July, 30 August, 30 September, and 20 October 2021. The seeds were removed from the fruits and immediately frozen in liquid nitrogen. After returning to the laboratory, the samples were stored at −80 °C.

### 4.6. RNA Extraction and RT-qPCR Analysis

The frozen stored seeds were ground into fine powder in liquid nitrogen. Approximately 140–170 mg powder samples were used to extract total RNA using the M5 HiPer Plant RNeasy Complex mini kit (Mei5 Biotechnology, Co., Ltd., Beijing, China). First strand cDNA was synthesized from 2 μg of total RNA using a Goldenstar™ RT6 cDNA synthesis Kit Ver.2 (TsingKe Biotech Co., Ltd., Beijing, China) according to the manufacturer’s instructions. One microliter of cDNA was used as a template for qPCR analysis using 2× TSINGKE^®^ Master qPCR Mix (SYBR Green I) (TsingKe Biotech Co., Ltd., Beijing, China). Analysis was performed on a LightCycler 96 Real-Time PCR System (Roche, Basel, Switzerland) and followed the program proposed by 2× TSINGKE^®^ Master qPCR Mix (SYBR Green I) protocol with a 56 °C annealing temperature and 45 cycles. The relative transcript level of each gene was normalized to glyceraldehyde-3-phosphate dehydro-genase gene (*GAPDH*) and calculated according to the 2^−ΔΔCt^ method. The relative expression of each gene on 30 June was set to 1, and those of all other stages were calculated relative to that of 30 June. Data represent the mean ± SD of three biological replicates. The primers used in this study are listed in Appendix A. The correlations between candidate genes and *bAS* were tested using the Pearson coefficient analyzed by two-sided tests using IBM SPSS 19 (SPSS Inc., Chicago, IL, USA).

## 5. Conclusions

In this study, we identified the members of multiple gene families that cover the whole triterpenoid backbone biosynthetic pathway, as well as CYP450 and UGT families, through mining the protein database for *C. oleifera* (Huashuo) by searching *HMM*. In total, 143 triterpenoid backbone biosynthetic genes, 1169 P450s, and 1019 UGTs from *C. oleifera* were identified. The CYP450s and UGTs were further categorized using tree-based methods. The transcriptome data analysis indicated that MVA was the main pathway for triterpenoid backbone biosynthesis in *C. oleifera*, and that HMGR and SE genes may play important roles in this pathway. Through WGCNA and RT-qPCR analysis, 11 *CYP450s*, 14 *UGTs*, and 8 TFs were identified as the candidate genes involved in theasaponin biosynthesis in *Camellia oleifera* seeds. The results of this study provide valuable information for further research investigating the biosynthesis and regulatory network of theasaponins.

## Figures and Tables

**Figure 1 ijms-23-06393-f001:**
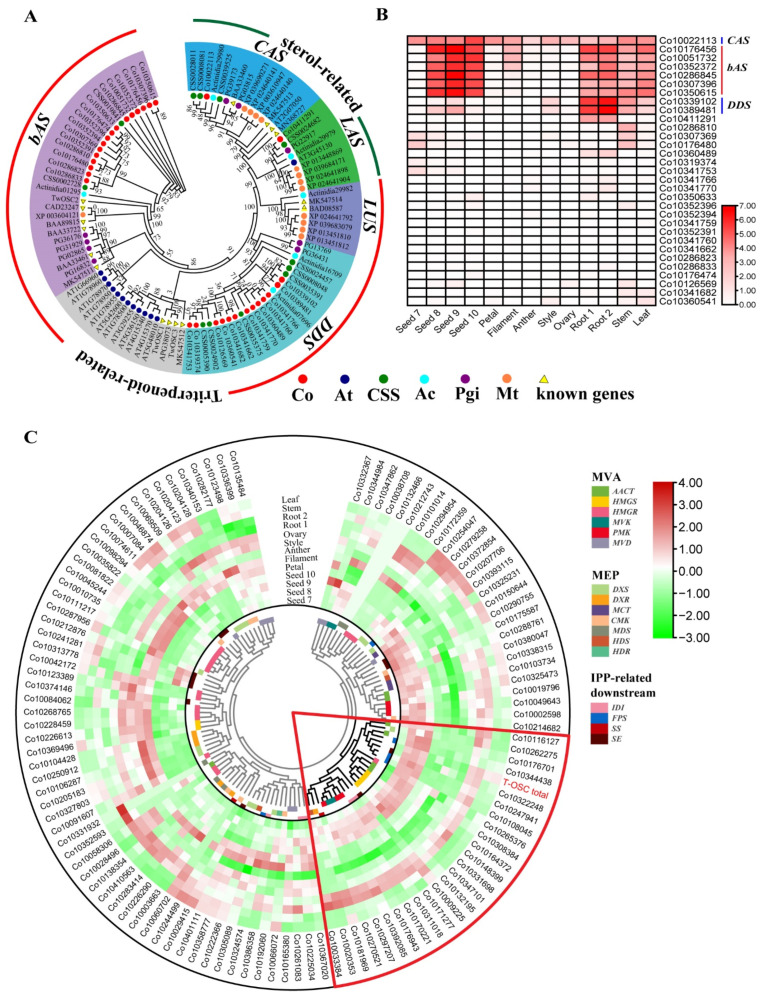
Analysis of triterpenoid backbone biosynthetic genes in *C. oleifera*. (**A**) Phylogenetic analysis of the OSC family. An approximate maximum likelihood tree was constructed by FastTree (v2.1.7 SSE3) using the JTT + CAT model and SH-like test with 1000 resamples. (**B**) Expression profiles of *OSCs* in different tissues and seed developmental stages. Gene expression is presented as log_2_(FPKM + 1). (**C**) Expression pattern hierarchical clustering for triterpenoid backbone biosynthesis genes based on RNA-Seq results. Seed 7, seed picked on 15 July; Seed 8, seed picked on 15 August; Seed 9, seed picked on 15 September; Seed 10, seed picked on 15 October; Root 1, root of containerized seedlings; Root 2, root of bare-root seedlings; known genes, previously characterized genes from other species; T-OSC total, the total FPKM values of triterpenoid-related *OSC* genes.

**Figure 2 ijms-23-06393-f002:**
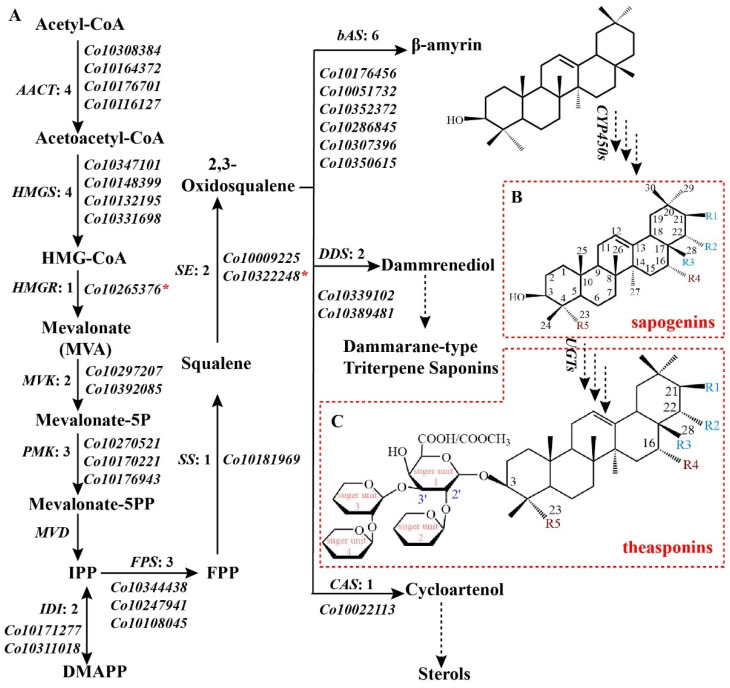
Putative biosynthesis pathway of theasaponin in *C. oleifera.* (**A**) Possible biosynthesis pathway of the triterpenoid backbones, with the candidate genes identified herein. Red asterisks (*) indicate genes that may play important roles in this pathway. (**B**) Basic structure of sapogenins from *C. oleifera* seeds. (**C**) Basic structure of theasaponins from *C. oleifera* seeds. R1-5 represent various modifying groups.

**Figure 3 ijms-23-06393-f003:**
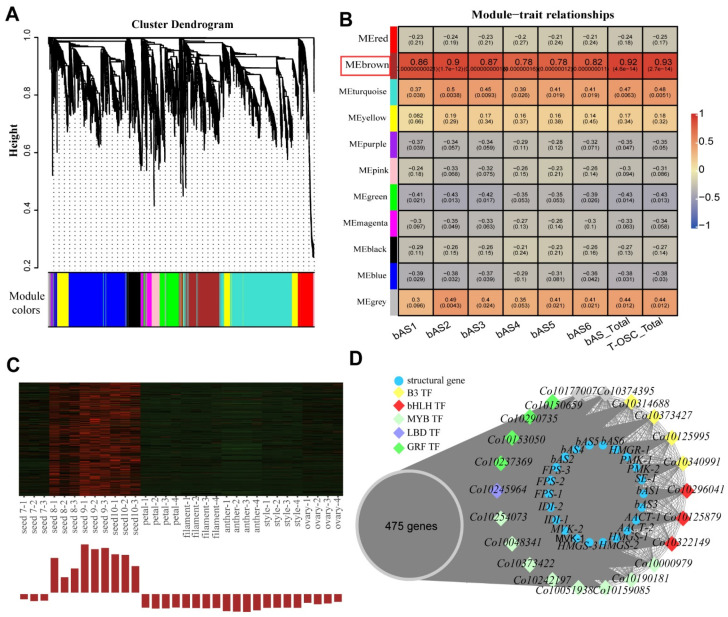
WGCNA identification of candidate genes related to theasaponin synthesis in *C. oleifera* seeds. (**A**) Cluster dendrogram and module assignment obtained by clustering the dissimilarity based on consensus topological overlap. (**B**) Module–trait relationships. Each row corresponds to a module. Each column corresponds to the expression patterns of *OSCs*. *bAS1*, *Co10051732*; *bAS2*, *Co10350615*; *bAS3*, *Co10307396*; *bAS4*, *Co10286845*; *bAS5*, *Co10176456*; *bAS6*, *Co10352372*. *bAS*-total, the total FPKMvalues of the six *bASs*. T-OSC-total, the total FPKM values of triterpenoid-related *OSC* genes. Each cell is colored by correlation according to the color legend. The number on the first line indicates the correlation, and the second line indicates the *P*-value in each cell. The MEbrown module with the highest correlation is indicated by the red box. (**C**) Heatmap of genes in the MEbrown module. Gene expression-level data in different tissues and seed developmental stages. Each sample comprised three or four replicates. (**D**) Network analysis of triterpenoid backbone biosynthetic genes and transcription factors in the MEblack module.

**Figure 4 ijms-23-06393-f004:**
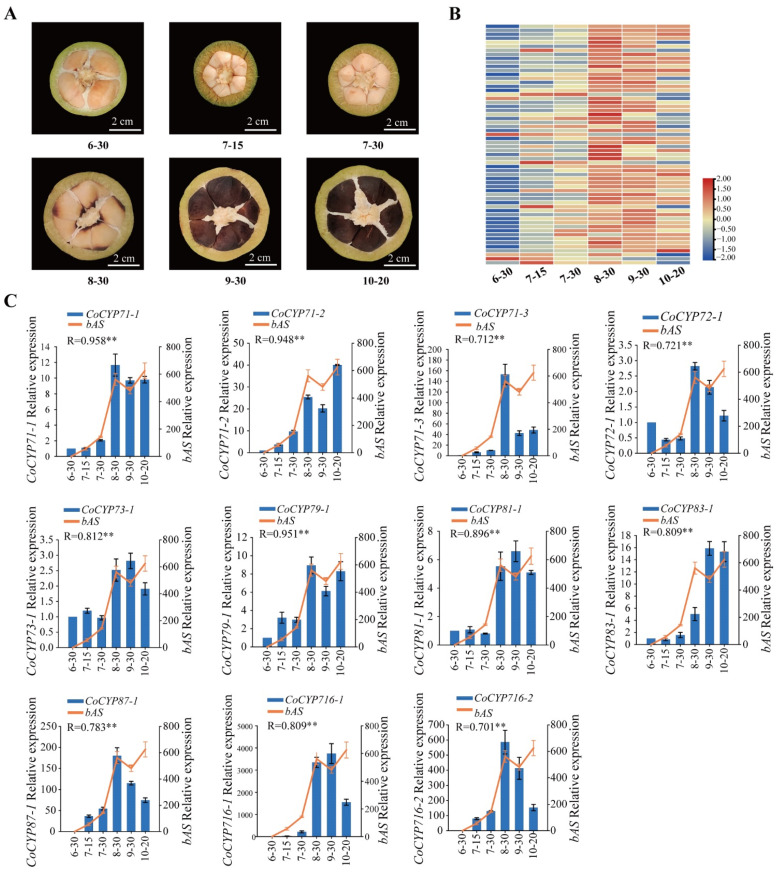
(**A**) *C. oleifera* seeds in different developmental stages. (**B**) Heatmap showing the relative expression levels of *CYP450s*, *UGTs*, and TFs screened by WGCNA. (**C**) The relative expression patterns of candidate *CYP450s* and *bAS*. Image 6-30, seeds picked on 30 June; 7-15, seeds picked on 15 July; 7-30, seeds picked on 30 July; 8-30, seeds picked on 30 August; 9-30, seeds picked on 30 September; 10-20, seeds picked on 20 October. The relative expression level of each gene was normalized to the glyceraldehyde-3-phosphate dehydrogenase gene (*GAPDH*) and calculated according to the 2^−ΔΔCt^ method. The relative expression level of each gene on 30 June was set to 1. Data represent the mean ± SD of three biological replicates. R represents the Pearson coefficient between candidate genes and *bAS*. ** *p* < 0.01 (two-sided test). The same annotations apply to Figure 5 and Figure 6 in this article.

**Figure 5 ijms-23-06393-f005:**
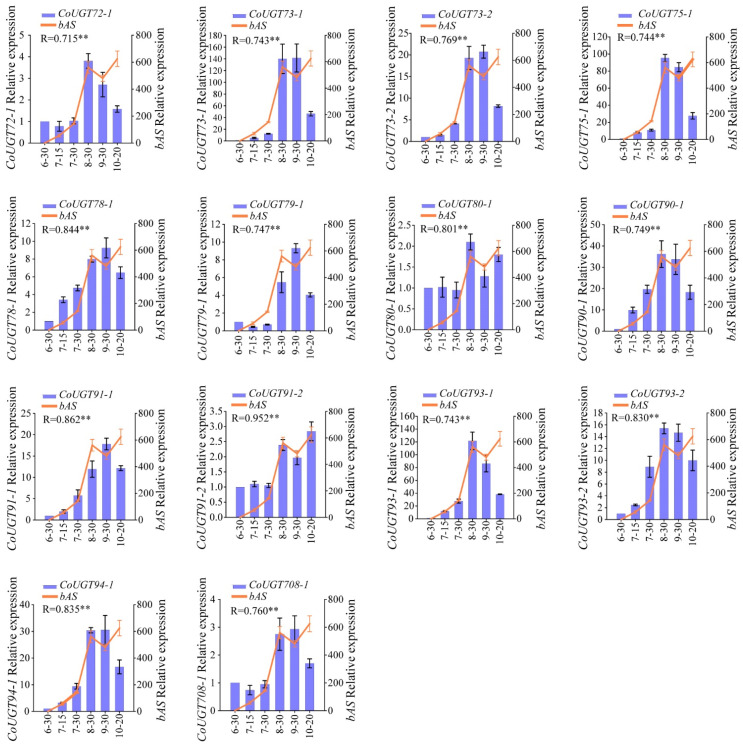
The relative expression patterns of candidate *UGTs* and *bASs*. ** *p* < 0.01 (two-sided test).

**Figure 6 ijms-23-06393-f006:**
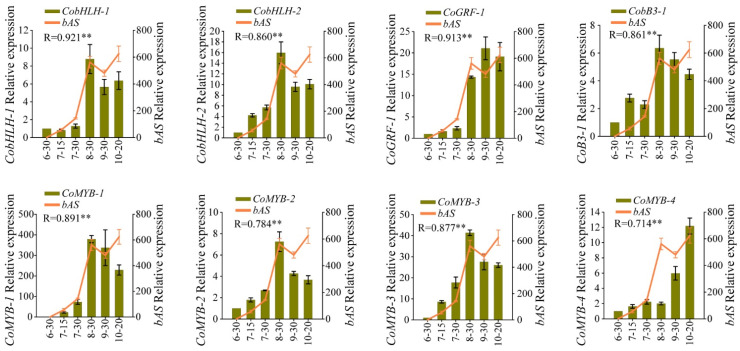
The relative expression patterns of candidate TFs and *bASs*. ** *p* < 0.01 (two-sided test).

**Table 1 ijms-23-06393-t001:** Statistics of triterpenoid biosynthesis genes.

Pathways	Families	Co	At	CSS	Ac	Pgi	Mt
MVA	AACT	7	2	6	4	5	4
	HMGS	6	1	2	2	5	2
	HMGR	18	2	5	5	8	11
	MVK	4	1	1	3	2	1
	PMK	6	1	3	5	4	1
	MVD	9	2	3	3	2	1
MEP	DXS	13	3	7	4	2	5
	DXR	6	1	1	1	5	2
	MCT	3	1	1	2	1	2
	CMK	8	1	2	1	0	1
	MDS	5	1	1	2	4	1
	HDS	3	1	3	2	5	1
	HDR	1	1	0	1	6	2
IPP isomerase	IDI	5	2	2	3	2	1
IPP-related downstream	FPS	5	2	4	4	5	1
	SS	2	2	2	2	4	1
	SE	11	6	7	6	20	10
OSC	Total	31	14	12	6	7	13
	Sterol-related	2	2	4	2	1	8
	Triterpenoid-related	29	12	8	4	6	5
	CYP450	1169	249	434	212	460	385
	UTG	1019	115	306	73	198	275

Co, *Camellia oleifera*; At, *Arabidopsis thaliana*; CSS, *Camellia sinensis*; Ac, *Actinidia chinensis*; Pgi, *Panax ginseng*; Mt, *Medicago truncatula.* The same annotations apply to all tables and figures in this article.

**Table 2 ijms-23-06393-t002:** Statistics of CYP450 Clans.

Clans	Families	Co	At	CSS	Ac	Pgi	Mt
CYP51	CYP51	5	1	1	1	4	2
CYP71	CYP701	9	1	2	3	4	1
	CYP703	6	1	1	1	2	1
	CYP705	0	25	0	0	0	1
	CYP706	38	7	16	2	11	2
	CYP71	134	50	57	8	53	71
	CYP712	0	2	0	0	6	1
	CYP73	12	1	4	0	4	2
	CYP75	33	1	15	2	5	9
	CYP76	67	8	41	7	22	27
	CYP77	9	5	3	3	5	2
	CYP78	40	6	10	9	7	4
	CYP79	55	10	1	3	6	7
	CYP81	72	18	32	10	16	10
	CYP82	102	5	28	9	20	19
	CYP83	21	2	2	1	12	19
	CYP84	12	2	6	6	48	19
	CYP89	16	7	6	5	9	11
	CYP93	6	1	3	0	2	20
	CYP98	9	3	4	2	4	1
CYP72	CYP709	0	3	0	0	0	1
	CYP714	34	2	11	8	5	9
	CYP715	2	1	2	4	0	6
	CYP72	97	9	39	15	46	24
	CYP721	6	1	2	2	7	2
	CYP734	3	1	4	3	5	1
	CYP735	7	2	2	4	4	2
CYP74	CYP74	15	2	4	5	4	6
CYP85	CYP702	0	6	0	0	0	0
	CYP707	12	4	6	8	15	6
	CYP708	0	4	0	0	1	0
	CYP716	65	3	24	12	14	3
	CYP718	23	1	8	7	5	4
	CYP722	6	1	3	4	2	3
	CYP724	5	1	3	2	2	2
	CYP85	7	2	6	7	7	3
	CYP87	57	1	9	9	8	1
	CYP88	12	2	3	3	9	15
	CYP90	19	5	7	12	19	10
CYP86	CYP704	19	3	10	6	11	9
	CYP86	15	11	5	7	7	8
	CYP94	54	6	27	8	24	10
	CYP96	34	13	18	3	12	23
CYP97	CYP97	11	3	3	4	6	3
CYP710	CYP710	3	4	1	0	0	1
CYP711	CYP711	14	1	5	2	5	3
	Others	3	1	0	5	2	1

**Table 3 ijms-23-06393-t003:** Statistics of UGT Clans.

Clans	Families	Co	At	CSS	Ac	Pgi	Mt
A	UGT79	27	11	11	0	9	13
	UGT80	11	2	3	1	6	7
	UGT91	81	3	14	7	8	15
	UGT94	99	0	24	6	20	0
B	UGT89	42	4	16	4	5	4
C	UGT90	17	3	5	1	1	0
D	UGT73	110	13	48	0	16	63
E	UGT71	60	14	7	4	21	12
	UGT72	98	9	23	3	4	41
	UGT88	32	1	6	2	1	8
	UGT708	10	0	4	1	2	3
F	UGT78	16	4	7	3	5	2
G	UGT85	121	6	34	7	17	45
H	UGT76	8	21	1	4	18	6
I	UGT83	15	1	8	1	6	5
J	UGT87	22	2	5	2	3	9
K	UGT86	5	2	2	1	3	0
L	UGT74	76	7	28	1	20	12
	UGT75	61	4	14	5	4	5
	UGT84	5	6	2	1	2	14
M	UGT92	26	1	11	7	4	2
N	UGT82	1	1	0	1	1	1
O	UGT93	48	0	16	2	6	3
	UGT95	8	0	4	1	1	1
P	UGT709	20	0	13	8	14	4

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
