# Peer review of "Whole-Genome Identification and Analysis of Multiple Gene Families Reveal Candidate Genes for Theasaponin Biosynthesis in Camellia oleifera"

_ijms, 2022, doi:10.3390/ijms23126393_

Round 1

Reviewer 1 Report

The authors have characterized the biosynthetic pathway of the asaponins in Camellia oleifera, and identified a total of143 triterpenoid backbone biosynthetic genes, 1,169 CYP450s, and 1,019 UGTs in C. oleifera. The authors further evaluated the expression profiles of the triterpenoid backbone biosynthetic genes in different tissues and seed developmental stages of C. oleifera. Their results revealed MVA as the main pathway for triterpenoid backbone biosynthesis. They further identified the candidate genes for theasaponin biosynthesis by WGCNA and qRT-PCR analysis, including 11 CYP450s, 14 UGTs, and 8 transcription factors.

I believe that the authors have provide sufficient background, explained the methodologies well, presented the results appropriately, and concluded appropriately based on available data.

I have no major technological concerns but a couple of minor suggestions for the authors to consider if a revision is requested by the editor:

Figure 3: resolution needs to be improved

Figure 4: resolution needs to be improved; alternatively, the parts in figure 4 may be presented in three separate figures.

Reviewer 2 Report

Dear Authors,

Reviewer comments ijms-1759733

The manuscript entitled „Whole-genome identification and analysis of multiple gene families reveal candidate genes for theasaponin biosynthesis in Camellia oleifera“ represents a valuable complex study aimed at an investigation of triterpenoid biosynthetic genes and their expression profiles in various tissues determined by RNAseq approach using WGCNA analysis and verified by qRT-PCR. The study led to the identification of candidate genes involved in theasaponin biosynthesis in C. oleifera.

I can recommend the manuscript for publication in International Journal of Molecular Sciences.

I have only a few comments on the present version of the manuscript:

1/ Abbreviations list: The manuscript contains a relatively high number of gene names abbreviations and also plant names abbreviations. Although they are explained in when used for the first time, I could recommend the authors to add a separate Abbreviations list in the manuscript.

2/ In Figure 1A, C providing the results of the phylogenetic analyses of the genes, I think that appropriate statistics to both phylogenetic trees has to be added, i.e., a scale bar providing information on the branches length or numbers at nodes providing information on the probability of the position of the branching points in the phylogenetic tree. In Materials and methods, the authors wrote that they used JTT+CAT model and SH-like test with 1000 resamples for the phylogenetic tree construction. The authors have to add appropriate reference on the algorithms used, and also add the information on the algorithms used for the phylogenetic tree construction into Figure 1 legend.

3/ Formal comments on the text related to English language:

Line 133: Add a comma between the words „….60, 110, 76, and 99 members“ and „respectively.“

Line 249: Add the word „in“ following the word „involved“ in the statement „To further screen for candidate CYP450s, UGTs and TFs involved in theasaponin biosynthesis…“

Line 364: Correct the word form „gene“ to „genes“ in „those genes“, , not „those gene“ in the statement „…four CYP450s belonging to CYP73, CYP79, CYP81, and CYP83 were also co-expressed with bAS. This suggested that those genes may have catalytic activity…“

Final recommendation: Accept after a minor revision.
